# Elimination of Osteosarcoma by Necroptosis with Graphene Oxide-Associated Anti-HER2 Antibodies

**DOI:** 10.3390/ijms20184360

**Published:** 2019-09-05

**Authors:** Hongmei Xiao, Peter E. Jensen, Xinjian Chen

**Affiliations:** 1Institute of Reproductive & Stem Cell Engineering, Central South University, Changsha 410083, China; 2Department of Pathology, University of Utah School of Medicine, Salt Lake City, UT 84132, USA

**Keywords:** osteosarcoma, necroptosis, HER2, graphene oxide, trastuzumab

## Abstract

The prognosis for non-resectable or recurrent osteosarcoma (OS) remains poor. The finding that the majority of OS overexpress the protooncogene HER2 raises the possibility of using HER2 as a therapeutic target. However, clinical trials on the anti-HER2 antibody trastuzumab (TRA) in treating OS find no therapeutic benefit. HER2 overexpression in OS is not generally associated with gene amplification, with low-level expression regarded as HER2 “negative”, as per criteria used to classify breast cancer HER2 status. Nevertheless, active HER2-targeting approaches, such as virus-based HER2 vaccines or CAR-T cells have generated promising results. More recently, it has been found that the noncovalent association of TRA with nanomaterial graphene oxide (GO) generates stable TRA/GO complexes capable of rapidly killing OS cells. TRA/GO induces oxidative stress and strong HER2 signaling to elicit immediate degradation of both cIAP (cellular inhibitor of apoptosis protein) and caspase 8, leading to activation of necroptosis. This is an attractive mechanism of cancer cell death as chemo/apoptosis-resistant tumors may remain susceptible to necroptosis. In addition, necroptosis is potentially immunogenic to promote tumor immunity, as opposed to apoptosis that tends to silence tumor immunity. Currently, no established anticancer therapeutics are known to eliminate cancers by necroptosis. The aim of this article is to review the rationale and mechanisms of TRA/GO-mediated cytotoxicity.

## 1. Introduction

Osteosarcoma (OS) is the most common primary malignant tumor of the bone. It primarily affects children and adolescents, while also occurring in young adults and the elderly [1]. Classified into several histopathological types with different prognosis, conventional OS represents the most common type [2]. Typically, OS arises from the metaphyseal area of long bones, including the femur, tibia, humerus, or pelvis, and other bones. In rare cases, OS may arise as extraosseous tumors. The standard therapy for OS consists of a combination of surgery with chemotherapy before and after surgery. Surgery aims to completely remove the local tumor. Prior to the use of chemotherapy, the 5-year survival rate with conventional OS is below 20%, even after amputation [3]. Neoadjuvant chemotherapy (before surgery) reduces tumor size and eliminates micro-metastases [4]. The degree of tumor necrosis in the resected tumor after neoadjuvant chemotherapy predicts the treatment outcome. Chemotherapy often consists of highly toxic agents such as methotrexate, leucovorin, doxorubicin, cisplatin, ifosfamide, and etoposide. Intensifying chemotherapy increases risk for serious drug toxicity and secondary cancers but does not improve survival.

Implementation of chemotherapy has a significantly improved prognosis, increasing the 5-year survival rate to approximately 60–80%. The survival rate for patients with extremity tumors is better compared to patients with axial tumors, reflecting the importance of the complete resection of local tumors. Pathologic fractures significantly reduce the chance to survive.

In spite of relatively low mortality rates for patients with local disease, most patients with metastatic OS die, and only a small fraction (25%) survive to five years. The OS often reoccurs with lung metastases, and the outcome for recurrent OS is even worse - only 15% of patients survive five years.

The dismal prognosis of metastatic or recurrent OS has not further improved over the last two to 3 decades, and currently remains very poor [1]. In addition, the severe, life-threatening drug toxicity and complications associated with intensified chemotherapy continue to be a major serious threat to the patient wellbeing [5]. The common primary reason for therapy failure is chemoresistance, where metastatic/recurrent OS no longer responds to chemotherapy. Various mechanisms are involved in chemoresistance [6], including the exclusion of drug molecules from the intracellular space by overexpression of energy-dependent transporters that actively pump drug molecules out of the cells, or acceleration in drug degradation or metabolism [7]. Disruption of DNA synthesis/replication is a common mechanism of many anticancer drugs; cancer cells, however, may evolve mechanisms to continue DNA repair and DNA synthesis despite the presence of the drugs. Disrupted DNA synthesis activates apoptosis to kill cancer cells, but OS cells may become apoptosis-resistant by upregulating anti-apoptotic proteins such as BCL-2 or by mutating pro-apoptosis genes such as P53. Small molecule tyrosine kinase inhibitors (TKI) have emerged as major players in cancer targeted therapy. *In vitro* experiments show that small molecule tyrosine inhibitors can regulate OS cell biology [8]. While relatively less toxic than chemo-drugs, TKIs tend to rapidly lose their inhibitory activity due to mutation of targeted molecules, or mutation of molecules in the signaling pathways to allow cancer cells to continue to survive and proliferate even when the upstream kinase activity is inhibited [9]. Alteration in micro-RNA has also been shown to play a role in OS drug resistance [10]. Micro-RNA miR-21 reportedly contributes to the pathogenesis of OS and could serve as a biomarker and therapeutic target [11]. Cancer stem cells may also mediate drug resistance [6]. While studies have reported various approaches to circumvent these resistance mechanisms, it is unlikely that another new conventional chemo-drug would completely overcome all resistance mechanisms.

Immunotherapy has attracted much attention in cancer therapy. Innate immune cell-based therapy, or immuno-stimulant [mifamurtide] in conjunction with standard chemotherapy is reported to offer therapeutic benefit [12]. Immune-check point blockage (anti-PD1/PDL1 and anti-CTLA-4) may provide positive therapeutic impact [13,14,15]. Notably, targeted therapies directed against human epidermal growth factor receptor 2 (HER2) appear to be quite promising.

In this article, we review aspects of HER2 biology and HER2 expression in OS, deliberating the rationale and mechanism of cytotoxicity mediated by graphene oxide (GO)-associated anti-HER2 antibody trastuzumab (TRA).

## 2. The Human Epidermal Growth Factor Receptor 2 (HER2) Expression in OS

HER2 is an oncogenic transmembrane tyrosine kinase receptor that promotes cell growth and survival [16]. Distinct from the other three members of the HER family receptors, the HER2 extracellular region adopts a configuration resembling a ligand-activated state. As a result, HER2 can undergo spontaneous dimerization and activation in the absence of ligands when HER2 is highly overexpressed on the cell surface [17]. HER2 can dimerize with other HER2 molecules to form homodimers or with other members of the HER family proteins, such as HER3 and HER4, to form heterodimers, leading to activation of HER2 tyrosine kinase [18]. Phosphorylation of HER2 downstream proteins activates the PI-3K-AKT-mTOR pathway through activation of RAS–MAPK, which prevents apoptosis and promotes cell cycle progression (Figure 1).

HER2 is overexpressed in a large percentage (about 60%) of OS as well as many other cancer types in addition to breast and gastroesophageal carcinomas [19,20,21,22,23,24,25,26,27,28,29,30,31,32,33]. Despite the widespread overexpression of HER2, the US FDA has only approved anti-HER2 therapy with anti-HER2 antibodies (trastuzumab and pertuzumab) for treatment of HER2-positive (HER2^+^) breast and gastroesophageal carcinomas because the antibody (Ab) therapies have not been found to be effective in other cancers. The levels of HER2 expression in other cancer types are generally low compared to HER2^+^ breast carcinoma. The criteria for HER2-positive status in breast carcinoma are strict, requiring complete, homogeneous, circumferential, intense membrane staining (IHC 3+) in greater than 10% tumor cells by immunohistochemistry. HER2 IHC 2+ is regarded as equivocal, and the presence of HER2 gene amplification is then required to establish a HER2-positive status. IHC 2+ without gene amplification or IHC 1+ is regarded as HER2 negative [34]. HER2 gene amplification occurs in about 25% of breast carcinomas [35], similar to the percentage of HER2^+^ breast carcinomas by IHC [36]. It is believed that HER2-driven oncogenesis results from overexpression of HER2 proteins on the cell surface due to HER2 gene amplification. HER2 overexpression promotes HER2 homo- or heterodimerization, resulting in activation of HER2 tyrosine kinase. The sustained HER2 activation and downstream signaling constitute a strong oncogenic stimulus.

The anti-HER2 Ab trastuzumab (TRA) is a humanized monoclonal antibody (Ab) initially developed for the treatment of HER2^+^ breast cancer. TRA can inhibit HER2+ breast cancer cells. When used in conjunction with chemotherapy, it prolongs survival of patients with metastatic carcinoma compared to chemotherapy alone. TRA binds to the extracellular juxtamembrane portion of HER2, interfering with HER2 dimerization and activation of HER2 tyrosine kinase activity [37,38]. Several other mechanisms are also believed to be involved in the action of TRA, including increased HER2 endocytosis/degradation, and inhibition of the shedding of the HER2 extracellular domain. Antibody-dependent cytotoxicity (ADCC) is believed to be an important indirect therapeutic mechanism of TRA [37,38]. Because of the success with TRA therapy in breast carcinoma, there has been extensive interest in using TRA for the treatment of other cancers, including OS.

HER2 overexpression in OS was first reported over two decades ago [39,40]. The overexpression was established by various experimental techniques, including immunohistochemistry (IHC), western blots, as well as quantitative real-time RT-PCR to measure HER2 RNA transcripts in laser-microdissected OS tissue [41]. Some studies reported that HER2-overexpressing OS was associated with a worse prognosis. There are even inconsistent reports of cytoplasmic HER2 protein in the absence of HER2 RNA [42]. Nevertheless, given the information in the literature, it is apparent that about 40–60% of OS overexpress HER2, but the levels of expression are generally lower than in HER2+ breast carcinoma. In addition, there is often no HER2 gene amplification in OS [43]. Therefore, the levels, as well as mechanism of HER2 overexpression in OS, appear to differ from those in breast carcinoma.

The low levels of HER2 overexpression, in the absence of HER2 gene amplification, imply that HER2 may not play a significant role in the oncogenesis of OS. Therefore, inhibition of HER2 kinase activity with anti-HER2 Ab may not inhibit OS. Indeed, when OS cell lines were treated with TRA in vitro, no inhibition was observed [44]. A phase II clinical trial of TRA has also be carried out to treat HER2^+^ metastatic OS. Despite the addition of TRA to intensive chemotherapy, no therapeutic benefit was observed [45]. Therefore, neither direct nor indirect mechanisms of TRA therapeutic action worked in OS. In addition, intensive chemotherapy may abrogate antibody-dependent cytotoxicity (ADCC) due to chemo-induced neutropenia with a loss of lymphocytes and NK cells. Therefore, inhibition of HER2 kinase activity by TRA does not provide therapeutic benefit in OS.

Despite the negative results of the clinical trial, other HER2-targeting approaches have demonstrated promising results. Recombinant (attenuated) Listeria monocytogenes expressing HER2 was used to immunize canines with HER2^+^ OS. The immunization resulted in a HER2-specific IFNα response towards the HER2 intracellular region in the majority of animals. This, along with limb amputation or salvage surgery, significantly reduced the incidence of metastasis in animals. As a result, immunized animals survived longer compared to the historical controls treated with only surgery plus chemotherapy [46].

Chimeric antigen receptor T cells (CAR-T) have recently attracted intensive research interest in oncology. Studies have reported success in using HER2-specific CAR-T cells for OS. Culture of HER2^+^ OS cells with HER2-specific CAR-T cells expressing transgenic CD28-zeta domain stimulates strong type-1 T cell (Th1) responses, generating cytotoxicity against the target OS cells. Transfer of the CAR-T cells into mouse models harboring xenografted OS led to tumor regression [44]. Thus, when HER2 is used as a target for immunologic attack, rather than a target for passive inhibition by HER2 antibody, HER2-targeted therapy may be effective in OS.

## 3. Association of Trastuzumab (TRA) with Graphene Oxide (GO]

To transform the passive inhibitory function of the anti-HER2 Ab into active cytotoxic activity, a TRA-toxin conjugate has been produced. The product trastuzumab emtansine (T-DM1) is TRA linked to a cytotoxic microtubule-inhibitory agent, DM1 [47,48]. The US FDA approved T-DM1 for the treatment of HER2-positive breast cancer with metastasis. Chemotherapy, in conjunction with T-DM1, is less toxic when compared to lapatinib plus capecitabine and is associated with longer survival of patients previously treated with TRA plus taxane. Despite encouraging results with breast carcinoma, no therapeutic benefit was observed of T-DM1 in the treatment of other cancer types, including pancreatic ductal carcinoma, non-small cell lung cancer (NSCLC), or OS, which generally express HER2 at lower levels (IHC1 1-2+) than HER2^+^ breast cancer [45,49,50,51,52]. In line with this result, we also find that T-DM1 manifested no cytotoxic activity towards HER2^+^ OS cell lines in vitro [53]. The lack of activity is most likely due to a lack of internalization of T-DM1 by OS cells because internalization requires higher levels of HER2 on the cell surface [54]. As a result, the toxin is not delivered into the cells—a key requirement for T-DM1 action. Therefore, in order to make anti-HER2 Ab-drug conjugates (ADC) cytotoxic to HER2^lo^ OS cells, TRA-drug conjugates need high avidity and increased binding capacity for HER2.

### 3.1. TRA Can Stably Associate with Graphene Oxide (GO) through Noncovalent Bonds

In previous studies, we reported that monoclonal antibodies (mAbs) can stably bind to nanomaterial graphene oxide (GO] through noncovalent bonds, generating GO-associated Ab complexes (Ab/GO] with high avidity for specific antigens. In addition, the antibody-GO complex (Ab/GO] is cytotoxic to target cells [55]. GO is the only natural material that is two-dimensional. Being one-atom-thick, GO has large planar surfaces that are extremely flexible and capable of binding multiple drug or protein/antibody molecules [56,57,58]. There has been intense research interest in GO for drug delivery [59,60]. Conventionally, GO is generated by oxidation of graphite using Hummer’s method. However, the Hummer’s method-produced GO is too large in size (200–2000 nm) for drug delivery. We find that intensive sonication with a probe sonicator can break GO nanosheets apart to approximately 100 nanometers. When incubated together in low salt solutions, GO, and antibody molecules spontaneously bind to each other through noncovalent interactions. To generate the trastuzumab (TRA)/GO complex (TRA/GO), TRA is mixed with GO at a 5 to 1 weight ratio in 10% PBS at 37 °C under constant agitation for 4–8 h. This results in the formation of highly stable TRA/GO complexes of approximately 1000 nm [53]. Each microgram of GO bonds approximately 5 μg of TRA. The association between GO and TRA is so stable that the TRA/GO complex does not dissociate even after multi-rounds of vigorous washing in PBS.

### 3.2. GO-Associated TRA Demonstrated Enhances Binding to HER2^lo^ OS Cells

There is a possibility that random, noncovalent association of TRA with GO blocks the Ag-binding sites of TRA and thus obliterates its reactivity with HER2. Interestingly, this does not seem to happen. When TRA/GO complexes were made with Fluorescein (FITC)-conjugated TRA to stain OS cells, it stained the HER2^lo^ OS cells extremely brightly, 50 to 100-fold as bright when compared to free FITC-TRA that only produced faint staining on the OS cells due to very low levels of HER2 expression [44]. FITC-TRA/GO did not stain human peripheral blood mononuclear cells (PBMCs) or lymphocytes that do not express HER2, suggesting that the bright staining on OS cells by FITC-TRA/GO is HER2-specific. Alternatively, FITC-anti-CD8/GO did not stain OS cells. Thus, the strong binding (as indicated by bright staining) of TRA/GO to OS cells is mediated by HER2-specific reactivity, and the random association of TRA with GO does not abolish but rather augments TRA reactivity with HER2. While it is unclear why the random association of TRA with GO does not disrupt HER2 reactivity of TRA, it is possible that TRA interacts with GO primarily through the Fc region of the antibody. Further experiments are needed to study GO binding activity of Fc vs. Fab/F(ab)2’ of the antibody. The enhancement in HER2 binding mostly results from an increase in valency of TRA due to the association of multiple TRA molecules with each GO nanosheet. The increase in valency is expected to increase Ab avidity. Indeed, TRA/GO demonstrates much higher avidity for HER2 as compared to free TRA in competition binding assays. TRA/GO is able to replace HER2-bound TRA on OS cells even when free TRA is present in hundreds fold higher concentrations compared to the concentrations of TRA/GO. Under microscopy, the TRA/GO bound to HER2 on OS cells is visualized as coarse aggregates (capping), contrasting to the binding of free TRA that appears as a fine homogeneous distribution along the cell membrane. The result of HER2 capping with TRA/GO further confirms a multi-valent nature of GO-associated TRA, as antibody-mediated capping results from ligand crosslinking by multivalent Abs [61].

### 3.3. TRA/GO Is Cytotoxic to OS Cells

In previous studies by others and a study of our own, free TRA showed no inhibitory effect on HER2^+^ OS cells in culture, even at very high concentrations [44]. In contrast, when the HER2^+^ OS cell lines—including MG63, HOS, and 143B, which are known to express only low levels of HER2—were exposed to TRA/GO in culture, the OS cells die within 24 h [53]. The death of the OS cells was determined by light microscopy using trypan blue or fluorescent LIVE/DEAD Dye. Flow cytometry, in conjunction with the LIVE/DEAD Dye, was also used to assess OS cell death. Under electron microscopy, the TRA/GO-killed OS cells manifested features of necrosis rather than apoptosis (see below). GO alone did not induce cell death. The culture media used were heat-inactivated to eliminate complement activity. Serum-free medium was also used in a previous study, and the results showed that complement was not required for the killing of target cells [Ref. 55]. Therefore, the TRA/GO-derived cytotoxicity is direct rather than complement-mediated. In more recent experiments, we tested TRA/GO on a panel of breast carcinoma cell lines expressing different levels of HER2, including MDA-MA-468, which is known not to overexpress HER2. TRA/GO manifested no cytotoxicity toward MDA-MA-468, while it killed other HER2^+^ cells including MCF-7 and JIMT-1 (unpublished data), further confirming that TRA/GO-mediated cytotoxicity is HER2-specific. It took up to three days for doxorubicin plus oxaliplatin, the chemotherapy drugs commonly used in OS treatment, to kill the target cells, in contrast to 24 h with TRA/GO. Therefore, TRA/GO-mediated cytotoxicity proceeds with very fast kinetics. TRA/GO did not affect the viability of human lymphocytes at concentrations that kill OS cells. Contrasted to the toxicity of chemo-drugs, which is nonspecific, doxorubicin plus oxaliplatin kills human lymphocytes at the minimum concentrations required to kill the OS cells.

### 3.4. TRA/GO Induces Oxidative Stress as well as HER2 Signaling in OS Cells

To understand the mechanism of TRA/GO mediated cytotoxicity, we examined the production of reactive oxygen species (ROS) in OS cells. At a low concentration (5 μg/mL), GO does not induce ROS, consistent with a previous report that only high (>50 μg/mL) concentrations of GO elicit oxidative stress [62]. Low concentration of TRA/GO (25/5 μg/mL), however, induces strong ROS production, demonstrating that, in the form of TRA/GO, low concentrations of GO are sufficient for ROS induction. The ROS-scavenger, Tiron, that extinguishes ROS [63], abrogates the killing of the OS cells by TRA/GO, indicating that ROS is required for cytotoxicity. However, when OS cells were treated with high concentrations (50 μg/mL] of free GO, or low concentration of GO (5 μg/mL) associated with a different mAb, W6.32, which recognizes MHC class I, robust production of ROS occurred, but no OS cell death was observed. This suggests that ROS is insufficient for the cytotoxicity, and HER2 signaling is also required for cytotoxicity. Lapatinib is a small molecule inhibitor of EGFR/HER2, which blocks HER2 signaling (Figure 1). When OS cells are treated with TRA/GO in the presence of lapatinib, no cell death occurs, despite strong ROS production. Therefore, HER2 signaling is indeed required in the cytotoxicity. Subsequent experiments demonstrated that TRA/GO binding to OS cells activates HER2 downstream pathways, including the phosphorylation of AKT and ERK. Therefore, the mechanism of action of TRA/GO is distinct from that of free TRA: whereas binding of TRA to HER2 inhibits HER2 tyrosine kinase activity and attenuates HER2 signaling, the binding of TRA/GO, in contrast, induces ROS production and augments HER2 signaling. TRA/GO-induced HER2 signaling most likely results from the crosslinking/aggregation (capping) of HER2 molecules as a result of TRA/GO binding.

### 3.5. TRA/GO Kills OS by Necroptosis

TRA/GO-killed OS cells are morphologically necrotic, characterized by ruptured plasma membranes, swollen cytoplasm, and nuclear lysis, distinct from apoptosis. Caspase 3 or PARP, key elements of apoptosis, are not activated, and there is no apoptotic DNA fragmentation. In line with this, TRA/GO-induced OS cell death cannot be stopped by the pan-caspase inhibitor, z-VAD-FMK. Therefore, TRA/GO-mediated killing is through cell death mechanisms other than apoptosis. Necroptosis is a recently recognized, caspase-independent, programmed necrosis, distinct from conventional necrosis, which is unprogrammed [64,65]. Activation of the necroptosis pathway requires the participation of the receptor-interacting protein kinase 1 (RIP1). Previous studies reported that the small molecule inhibitor, necrostatin 1 (Nec-1), can inhibit necroptosis by allosterically inhibiting RIP1. When added to cell culture, Nec-1 prevented OS cell death by TRA/GO [53], suggesting that TRA/GO kills OS cells by necroptosis. The mixed lineage kinase domain-like protein (MLKL) is the membrane disruption unit in necroptosis [66]. The inhibitor of MLKL also abrogated the cytotoxicity of TRA/GO [53], further indicating that necroptosis is the mechanism of TRA/GO-mediated killing of OS cells. Necroptosis occurs in viral infection as host cell caspase activity is inhibited by virus-derived caspase inhibitors. Necroptosis is also involved in a variety of disease processes, including myocardial infarction, inflammatory bowel disease, pancreatitis, psoriasis, rheumatoid arthritis, and neuronal death in degenerative neurological disorders. Little is yet known of necroptosis in cancers. While necroptosis is classified into three categories, 1) extrinsic, elicited by TNFα; 2) intrinsic, triggered by reactive oxygen species (ROS); and 3) ischemic, only the TNFα/TNFR1-mediated necroptosis is relatively well characterized. The in vitro model of TNFα-mediated necroptosis involves activation of TNFα receptor (TNFR1) in the presence of exogenous caspase inhibitor [67]. TNFR1 binding protein TNFR-associated death protein, TRADD, and TNF receptor-associated factor 2, TRAF2, activate RIP1 to recruit RIP3 to form RIP1/RIP3 necrosome or ripoptosome. The execution of necroptosis commonly involves the formation of the RIP1/RIP3/MLKL complex [68,69]. When lysates of OS cells treated with TRA/GO were immunoprecipitated (IP) with an anti-RIP3 antibody, both RIP1 and MLKL were co-precipitated with RIP3, demonstrating formation of RIP1/RIP3/MLKL complexes. Therefore, TRA/GO kills OS cells by necroptosis, an unprecedented mechanism of OS cell death by therapeutics. To further understand the molecular mechanism of how TRA/GO activates necroptosis, we examined the status of the cellular inhibitor of apoptosis protein (cIAP) in OS cells upon TRA/GO treatment. cIAP is an important intracellular protein regulating the early events of the common pathway for programmed cell death. Loss of cIAP may trigger cell death by apoptosis or necroptosis, depending on the presence or absence of caspase 8 activity [70]. Mitochondria damage is often associated with oxidative stress [71], and the release of second mitochondria-derived activator of caspases (Smac/DIABLO) can lead to degradation of cIAPs [72,73]. Loss of cIAP promotes the formation of the intracellular platform complexes containing RIP1 and FADD. In the presence of caspase 8, the complex activates the apoptosis cascade [74]. In the absence of caspase 8 activity, however, the cell death program promotes the formation of the RP1/RP3/MLKL complex and proceeds to necroptosis [75,76]. Remarkably, treating OS cells with TRA/GO results in immediate loss of cIAP within 5 min, suggesting that depletion of cIAP might act as an important early event in activation of OS cell death program. Surprisingly, in addition to cIAP, caspase 8 is also lost instantly after TRA/GO treatment. Thus, TRA/GO provokes rapid depletion of both cIAPs and caspase 8 in OS cells. The simultaneous loss of caspase 8 and cIAP explains the unnecessity for exogenous caspase inhibitors to inhibit caspase 8 for necroptosis to occur. The rapid kinetics, at which cIAP and caspase 8 are degraded, and the RIP1/RIP3/MLKL complex is formed upon TRA/GO treatment, explain the fast speed by which TRA/GO kills OS cells. No other therapeutic agents are currently known to kill OS by necroptosis. Therefore, the findings are novel.

### 3.6. TRA/GO Eliminate Established Xenograft OS

The in vivo therapeutic efficacy TRA/GO was studied in a xenograft OS mouse model carrying established subcutaneous tumors derived from human OS cell lines [77]. Treatment was not started until solid tumors grew to approximate 40 mm^3^. Intravenous treatment with TRA/GO stopped tumor growth and was followed by subsequent tumor shrinkage, whereas tumors in the other treatment groups grew rapidly. Pathological examination identified no tumor in any TRA/GO-treated mice at the sites of initial tumor growth or OS cell inoculation. Occasional minute (1 to 3 mm) benign-appearing lesions were identified in some TRA/GO-treated mice. Microscopic examination of these lesions identified granulomas, suggesting clearance of tumor necrosis by scavenger activity. In separate experiments, OS cells were transplanted intravenously to induce disseminated OS in the lung, a major poor prognosis indicator in patients. Treatment with intravenous TRA/GO also eliminated established OS in the lung, resulting in the indefinite survival of the animals, while all the mice in the other treatment groups succumbed to lung OS [53].

## 4. Discussion and Conclusions

HER2 may represent a significant therapeutic target in OS as approximately 60% of OS overexpress HER2. While the low levels of HER2 on OS cells are insufficient for inhibitory therapy by anti-HER2 antibodies, low levels appear to be sufficient targets for cytotoxic therapies. In the light of other active HER2-targeted therapies such as the virus-based HER2 immunization or HER2 CAR-T cells, TRA/GO therapy appears safe, effective, and easy to institute. The cell death pathway, necroptosis, by which TRA/GO eliminates OS cells, is unprecedented. How TRA/GO-induced ROS production and HER2 signaling give rise to necroptosis remains to be elucidated. Our data and information available in the literature [72,73,78,79,80,81,82,83,84,85,86] allow us to hypothesize the following (Figure 2). The leak of Smac/DIABLO as a result of oxidative stress-induced mitochondria damage promotes cIAPs degradation. HER2-signaling is required for activation of NF-***κ***B and TNFα production as well as caspase 8 degradation [86,87]. Depletion of cIAPs results in the formation of the death complex ripoptosome [88]. The loss of caspase 8 allows the formation of RIP1/RIP3/MLKL complexes and execution of necroptosis (Figure 2).

Much remains to be learned with regard to the mechanism of TRA/GO-mediated cytotoxicity. It is not yet clear whether TRA/GO is internalized, what pathway is responsible for GO- and TRA/GO induced production of ROS, and whether or not autocrine TNFα production plays any role in necroptosis. Further studies are needed.

Eradication of established OS in vivo by intravenous TRA/GO is remarkable, given that OS cells are insensitive to TRA [44]. Only cancers expressing IHC 3+ high levels of HER2 are known to respond to HER2 antibody therapy [89]. No established chemotherapy drugs are known to kill cancer cells by necroptosis, although studies report inhibition of cancer cell proliferation in vitro by high concentration GO [90]. Necroptosis is considered an attractive form of cancer death because chemo-resistant cancers may be susceptible to necroptosis [91]. This is because chemotherapy often kills cancer cells by apoptosis [92,93]. Constitutive activation of tyrosine kinases downstream of HER2-signaling in cancer cells represents one of the common mechanisms of TRA-resistance. Because TRA/GO kills OS by stimulating rather than by inhibiting HER2 signaling [38,94], it is likely that those TRA-resistant cancers may still be susceptible to TRA/GO [38]. Necroptosis tends to be immunogenic with potential to activate anti-tumor immunity [95]. The lack of toxicity to human lymphocytes offers a significant opportunity for the implementation of immunotherapy in combination with TRA/GO. Combination of targeted therapy with immunotherapy may represent one of the most effective approaches in cancer therapy. Given the fact that HER2 is overexpressed in a large variety of cancers, the TRA/GO formulation may constitute an effective therapy for many cancer types. In preliminary studies, TRA/GO demonstrates similarly potent cytotoxicity towards cell lines of breast, pancreatic, and lung cancers.

In summary, TRA can be stably associated with GO through non-covalent bonds, and GO-associated TRA has a much-enhanced binding capacity for HER2 on OS cells. Binding of TRA/GO eliminates OS cells by necroptosis. TRA/GO-mediated cytotoxicity has faster kinetics compared to chemo-drugs and does not affect human lymphocytes. TAG/GO induces both ROS and HER2 signaling, which activate a necroptosis pathway by depleting both cIAP and caspase 8. Intravenous administration of TRA/GO eliminates established xenograft OS without obvious side effects in mouse models.

## Figures and Tables

**Figure 1 ijms-20-04360-f001:**
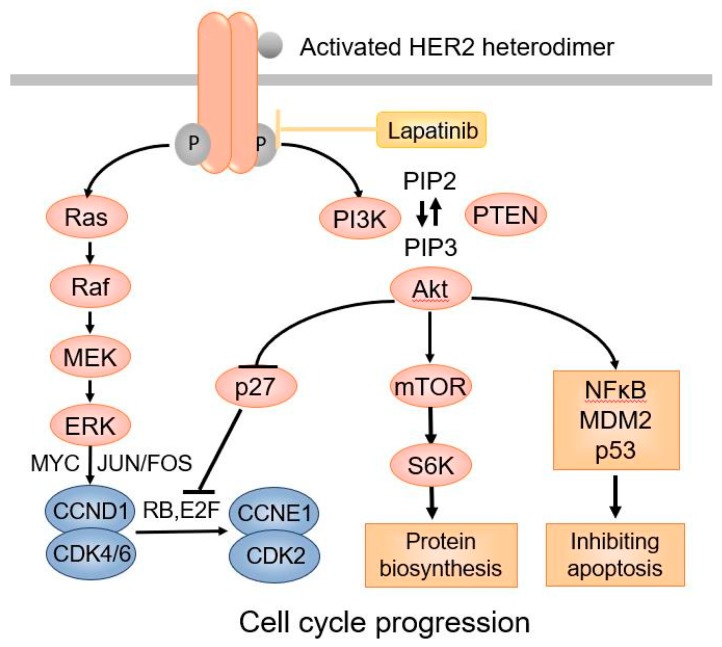
The HER2 signaling pathway.

**Figure 2 ijms-20-04360-f002:**
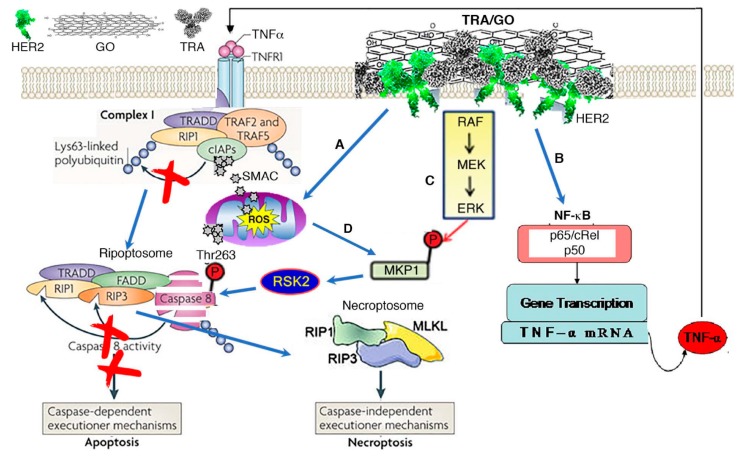
Mechanism of TRA/GO action. (**A**) TRA/GO induces ROS production. ROS disrupts mitochondrial integrity, releasing SMAC/DIBLO into cytosol. Binding of SAMC to cIAPs prevent RIP1 ubiquitylation, allowing formation of death complex ripoptosome. (**B**) Binding of TRA/GO to HER2 causes HER2 homodimerization, eliciting HER2 signaling; this activates NF-kB through the canonic pathway, augmenting TNFα production. TNFα singling through TNFR1 promotes degradation of cIAPs. (**C**) HER2 signaling also activates MAPK pathway, resulting in activation RSK2 that phosphorylate caspase 8 at Thr263 site to induce caspase ubiquitination and proteosomal degradation. Degradation of caspase 8 leads to formation of necroptosome executing necroptosis. (**D**) ROS also contributes to MAPK activation.

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
