# Peer review of "Elimination of Osteosarcoma by Necroptosis with Graphene Oxide-Associated Anti-HER2 Antibodies"

_ijms, 2019, doi:10.3390/ijms20184360_

Round 1

Reviewer 1 Report

The topic of the article review is of particular interest for clinicians and researchers in the field of osteosarcoma treatment. The current treatment (surgery and pre-post surgery chemotherapy) has not been significantly improved over the last two to 3 decades, and so far the 5 years survival rate is approximately around 60-80% worldwide. In addition, the severe, life-threatening drug toxicity and complications associated with intensified chemotherapy in order to completely eliminate cancer cells, continue to be a major and serious threat to the patient wellbeing, especially considering the low average age of affected patients. 

In particular, the authors aim to provide a detailed literature review on the usage of human epidermal growth factor receptor 2 (HER2) inhibitors such as the antibody trastuzumab  (TGA) as co-adjuvant chemotherapeutic agent alone or in combination with GO (Graphene oxide nanoparticles). Nevertheless, as the authors underline, the low levels of HER2 overexpression and the absence of HER2 gene amplification in the majority of OS subtypes,  implies that HER2 may not play a significant role in OS oncogenesis. This also makes, in my opinion, the rationale behind the usage of HER2 inhibitors for OS chemotherapeutic treatment questionable and controversial.  In addition recent clinical trial for OS treatment using TGA, have shown negative results, despite other HER2-targeting approaches have demonstrated promising results. Therefore, in order to make anti-HER2 Ab-drug conjugate cytotoxic to HER2-low expressing OS cells, TRA-drug conjugates need high avidity and increased binding capacity for HER2. One option is represented by the association of TGA with Graphene Oxide nanoparticles, which appear to be promising for the vehiculation of TGA in OS cells that poorly express the HER2 receptors. Despite the mechanism of the assumed increased binding capacity of TGA in presence of GO nanoparticles is poorly described and poorly known in literature, this targeting approach opens up a possibility to further investigate its molecular mechanis. Along the manuscript, the authors report relevant information regarding the possible mechanism of TRA/GO mediated cytotoxicity, but it would be important to add data or literature regarding the usage of TGA/GO in HER2 negative tumor cells, to further strength the direct involvement of TGA/GO towards HER2 inhibitory effect.

MAJOR REVISION: I suggest an extensive revision of the English grammar, syntax and of the several editing typos along the entire manuscript, before the re-submission.

Lane 51: “Various mechanisms are involved in chemoresistance, including exclusion of drug from inside of cells by overexpression of the energy-dependent transporters that actively pumps drug molecules outside of the cells, or acceleration in drug degradation or metabolism (6)”:  I suggest to cite more relevant references or article reviews in the field of OS treatment and mechanism of resistance.

Lane 80: a graphical abstract of HER2 downstream signalling pathway in physiological conditions would help the reader who is not familiar with HER2 mechanism of action.

Author Response

Reviewer 1:

Along the manuscript, the authors report relevant information regarding the possible mechanism of TRA/GO mediated cytotoxicity, but it would be important to add data or literature regarding the usage of TGA/GO in HER2 negative tumor cells, to further strength the direct involvement of TGA/GO towards HER2 inhibitory effect.

Response: In recent experiments, we have tested TRA/GO on a panel of breast carcinoma cell lines expressing different levels of HER2, and found that TRA/GO is not cytotoxic to the HER2-negative cell line MDA-MA-468 while it kills other HER2+ cell lines including MCF-7 and JIMT-1. This result is now described in the revised manuscript.

MAJOR REVISION: I suggest an extensive revision of the English grammar, syntax and of the several editing typos along the entire manuscript, before the re-submission.

Response: The manuscript has been edited by two native English-speaking scholars including the co-author Peter Jensen, and proofread.

Lane 51: “Various mechanisms are involved in chemoresistance, including exclusion of drug from inside of cells by overexpression of the energy-dependent transporters that actively pumps drug molecules outside of the cells, or acceleration in drug degradation or metabolism (6)”:  I suggest to cite more relevant references or article reviews in the field of OS treatment and mechanism of resistance.

Response: A recent review article is cited.

Lane 80: a graphical abstract of HER2 downstream signalling pathway in physiological conditions would help the reader who is not familiar with HER2 mechanism of action.

Response: This has been provided in the revised manuscript (diagram 1).

Reviewer 2 Report

This well-written manuscript describes the development and use of graphene-oxide associated transtuzumab (TRA-GO) in osteosarcoma treatment. The manuscript details briefly the current clinical prognosis of OS and the biology of HER-2, and explains why previous attempts to treat OS with TRA failed. It is built with a clear rationale, making it very easy to follow. However, less is detailed about GO and its physical and biochemical properties, and its effects on cells when it is not conjugated to antibodies.      

Major comments

Lines 183-185: the idea that GO binds to the Fc fragment of the antibody and thus increases its valency could be easily proven by using the Fab or F(ab)2' fragments of the antibody or svFc of the antibody: do these fragments bind GO as well? Has this been done? Lines 195-203: to prove that TRA-GO kills only Her2-positive cells, it should be incubated with tumor cell lines that are Her2-negative or with OS cell lines that are KO for Her2. Lymphocytes are not necessarily the best negative control. Has this been done? The in vitro experiments that show TRA/GO killing of tumor cells are crucial, as they provide isolated and controllable environment. In these experiments, was TRA/GO alone introduced to the cells? Were the cells incubated in complete medium that contains complement proteins that could drive complement-derived cytotoxicity (CDC)? This point is unclear. There are insufficient references, especially concerning the in vitro and

This well-written manuscript describes the development and use of graphene-oxide associated transtuzumab (TRA-GO) in osteosarcoma treatment. The manuscript details briefly the current clinical prognosis of OS and the biology of HER-2, and explains why previous attempts to treat OS with TRA failed. It is built with a clear rationale, making it very easy to follow. However, less is detailed about GO and its physical and biochemical properties, and its effects on cells when it is not conjugated to antibodies.      

Major comments

Lines 183-185: the idea that GO binds to the Fc fragment of the antibody and thus increases its valency could be easily proven by using the Fab or F(ab)2' fragments of the antibody or svFc of the antibody: do these fragments bind GO as well? Has this been done? Lines 195-203: to prove that TRA-GO kills only Her2-positive cells, it should be incubated with tumor cell lines that are Her2-negative or with OS cell lines that are KO for Her2. Lymphocytes are not necessarily the best negative control. Has this been done? The in vitro experiments that show TRA/GO killing of tumor cells are crucial, as they provide isolated and controllable environment. In these experiments, was TRA/GO alone introduced to the cells? Were the cells incubated in complete medium that contains complement proteins that could drive complement-derived cytotoxicity (CDC)? This point is unclear. There are insufficient references, especially concerning the in vitro and in vivo experiments with TRA/GO.    The concluding part should include more open questions and outline future prospects. For example, is TRA/GO internalized? What is the pathway responsible for GO-induced production of ROS? Is autocrine TNFa production necessary, or can necroptosis occur without it?

Minor comments: 

Line 66: 'sever' should be serve Minor English editing is needed. For example, in line 170, the word 'not' is missing. Line 206: are the GO concentrations provided true? Is 5 g/ml considered a low concentration?    Line 288: Greek symbols are missing (possible in other lines as well).

Author Response

This well-written manuscript describes the development and use of graphene-oxide associated trastuzumab (TRA-GO) in osteosarcoma treatment. The manuscript details briefly the current clinical prognosis of OS and the biology of HER-2, and explains why previous attempts to treat OS with TRA failed. It is built with a clear rationale, making it very easy to follow. However, less is detailed about GO and its physical and biochemical properties, and its effects on cells when it is not conjugated to antibodies.     

Major comments

Lines 183-185: the idea that GO binds to the Fc fragment of the antibody and thus increases its valency could be easily proven by using the Fab or F(ab)2' fragments of the antibody or svFc of the antibody: do these fragments bind GO as well? Has this been done? Lines 195-203: to prove that TRA-GO kills only Her2-positive cells, it should be incubated with tumor cell lines that are Her2-negative or with OS cell lines that are KO for Her2. Lymphocytes are not necessarily the best negative control. Has this been done? The in vitro experiments that show TRA/GO killing of tumor cells are crucial, as they provide isolated and controllable environment. In these experiments, was TRA/GO alone introduced to the cells? Were the cells incubated in complete medium that contains complement proteins that could drive complement-derived cytotoxicity (CDC)? This point is unclear. There are insufficient references, especially concerning the in vitro and in vivo experiments with TRA/GO.    The concluding part should include more open questions and outline future prospects. For example, is TRA/GO internalized? What is the pathway responsible for GO-induced production of ROS? Is autocrine TNFa production necessary, or can necroptosis occur without it?

Response: The authors very much appreciate the reviewer’s comments and suggestions!

The experiments suggested by the reviewer on comparing binding capacity of Fc vs Fab/F(ab)2' to GO are excellent and incorporated in the revised manuscript. We also agree with the reviewer that lymphocytes are not the best controls. In recent experiments, we tested TRA/GO on a panel of breast carcinoma cell lines expressing different levels of HER2, and found that TRA/GO is not cytotoxic to the HER2-negative cell line MDA-MA-468 while it kills other HER2+ cell lines including MCF-7 and JIMT-1. All the culture media used are heat-inactivated of complement. In addition, serum free medium was also tested in a previous publication (Ref. 55), and the results showed that complement is not required for the cytotoxicity. More reference information has been provided in the revised manuscript, and conclusion has been revised according to the reviewer’s suggestion to address open questions and future directions.

Minor comments: 

Line 66: 'sever' should be serve Minor English editing is needed. For example, in line 170, the word 'not' is missing. Line 206: are the GO concentrations provided true? Is 5 g/ml considered a low concentration?    Line 288: Greek symbols are missing (possible in other lines as well).

Response: The manuscript has been edited by two native English-speaking scholars including the co-author Peter Jensen, and proofread before resubmission. The low GO concentration is 5 micrograms/ml.

Reviewer 3 Report

The manuscript “Elimination of osteosarcoma by necroptosis with 2 graphene oxide-associated anti-HER2 antibodies” by Xiao et al. describes the ability of GO-associated Trastuzumab complex to induce osteosarcoma cells death by necroptosis highlighting the potential role of alternative therapeutic strategies to induce cell death in cancer resistant to treatment with antibodies or to cytotoxic agents.

There are some suggestions to the authors:

-The authors should underline with more evidences the aim of the present review in the Abstract section es well as in the Introduction.

- The authors should give more space in the text to describe the association of graphene oxide to antibodies and to the ability of these complex to be useful as an alternative therapeutic strategies in different types of cancers. Furthermore, it should be important for a better understanding of the proposed topic, give a more extensive description of the necroptosis pathway and its intracellular signaling mediators in cancer cells.

The references need to be improved with more recent papers on the presented topics.

Author Response

Reviewer 3:

The manuscript “Elimination of osteosarcoma by necroptosis with 2 graphene oxide-associated anti-HER2 antibodies” by Xiao et al. describes the ability of GO-associated Trastuzumab complex to induce osteosarcoma cells death by necroptosis highlighting the potential role of alternative therapeutic strategies to induce cell death in cancer resistant to treatment with antibodies or to cytotoxic agents.

There are some suggestions to the authors:

-The authors should underline with more evidences the aim of the present review in the Abstract section es well as in the Introduction.

Response:

The authors would like to thanks the reviewer for the helpful comments and suggestions.

The aim is now explicitly stated in the abstract and introduction.

The authors should give more space in the text to describe the association of graphene oxide to antibodies and to the ability of these complex to be useful as an alternative therapeutic strategies in different types of cancers.

Response: Text is now added to describe generation of TRA with GO complex, and the potential of TRA/GO to treat other cancer types.

Furthermore, it should be important for a better understanding of the proposed topic, give a more extensive description of the necroptosis pathway and its intracellular signaling mediators in cancer cells.

Response: Additional information is now provided on necroptosis pathway.

The references need to be improved with more recent papers on the presented topics.

Response: References are further updated.

Round 2

Reviewer 1 Report

The authors addressed all the raised concerns